# Morphological and Physiological Indicators for Screening Cell Lines with High Potential for Somatic Embryo Maturation at an Early Stage of Somatic Embryogenesis in Pinus Koraiensis

**DOI:** 10.3390/plants11141867

**Published:** 2022-07-18

**Authors:** Chunxue Peng, Fang Gao, Hao Wang, Iraida Nikolaevna Tretyakova, Alexander Mikhaylovich Nosov, Hailong Shen, Ling Yang

**Affiliations:** 1State Key Laboratory of Tree Genetics and Breeding, School of Forestry, Northeast Forestry University, Harbin 150040, China; chunxue@nefu.edu.cn (C.P.); fanggao@nefu.edu.cn (F.G.); 15636830205@163.com (H.W.); 2State Forestry and Grassland Administration Engineering Technology Research Center of Korean Pine, Harbin 150040, China; 3Laboratory of Forest Genetics and Breeding, Institution of the Russian Academy of Sciences, V.N. Sukachev Institute of Forest Siberian Branch of RAS, 660036 Krasnoyarsk, Russia; culture@ksc.krasn.ru; 4Department of Cell Biology, Institute of Plant Physiology, K.A. Timiryazev Russian Academy of Sciences, 127276 Moscow, Russia; al_nosov@mail.ru; 5Department of Plant Physiology, Biological Faculty, Lomonosov Moscow State University, 119991 Moscow, Russia

**Keywords:** *Pinus koraiensis*, embryonic cell proliferation, somatic embryo yield, morphological characteristics, physiological characteristics

## Abstract

Many cell lines in the embryogenic callus cannot produce somatic embryos (SEs) even if they meet the optimal SE maturation culture conditions during conifer somatic embryogenesis. This phenomenon hinders the progress of the industrial-scale reproduction of conifers. Therefore, there is an urgent need to obtain morphological and physiological markers to screen embryogenic calli in response to SE maturation conditions. To detect cell lines with high somatic embryogenesis potential during the proliferation process, we counted the number of pro-embryos and early SEs (ESEs) in different cell lines and storage substances, endogenous hormones, and polyamine contents. The results showed that the yield of *P. koraiensis* SEs was heavily dependent on genotype (*p* = 0.001). There were high levels of PE III (pro-embryo III) number, ESE number, and soluble protein content, in the response cell lines (R cell lines), which were 1.6-, 3-, and 1.1-fold those of the obstructive cell lines (B cell lines), respectively. The B cell line had high levels of starch, auxin (IAA), Put, Spd, and putrescine: spermine (Put: Spm) compared to the R cell line. In addition, the numbers of PE III, ESEs, and soluble protein content were significantly positively correlated with SE yield. In contrast, the contents of starch, abscisic acid (ABA), Put, Spm, and Spd were significantly negatively correlated with SE yield. To ensure the accuracy of the results, we used nine cell lines to test the results. The PE III and ESE numbers and the Spm and Spd contents were positively correlated with SE yield, while the levels of starch, ABA, IAA, Put: Spd, and Put: Spm were negatively correlated with SE yield. Thus, we recommend using high PE III and ESEs as morphological indicators and low levels of starch, IAA, ABA, and Put: Spm as physiological markers to screen cell lines with a high somatic embryogenesis potential. In addition, we also found that the relationship between Spd, Spm, and SE yield was opposite in the two experimental results. Therefore, we speculate that the differences in Spd and Spm content are mainly affected by genotype. In conclusion, this study obtained the morphological and physiological markers of some high-somatic embryogenic cell lines by comparing the differences between nine somatic embryogenic cell lines. Our results can guide the improvement of conifer somatic embryogenesis technology and can provide a theoretical basis for accelerating the application of biotechnology in large-scale artificial breeding.

## 1. Introduction

Somatic embryogenesis is an advanced propagation technology for higher plants. It is effective at producing a large number of high-quality seedlings quickly and is crucial for sustainable forestry, commercial forest development, and forest management [1,2]. However, in conifer somatic embryogenesis, problems, such as low embryonic tissue induction efficiency, low somatic embryo (SE) yield, low germination/conversion percentage, and slow growth of seedlings, limited the commercial development of this technology [3,4,5]. In an earlier study of *Pinus koraiensis* somatic embryogenesis, we discovered that numerous cell lines in the embryonic tissue were unable to generate SEs even if they met the ideal SE maturation culture conditions [6]. Such a phenomenon is also commonly seen in somatic embryogenesis of *Araucaria angustifolia*, *P. taeda*, and *Picea abies* [7,8,9]. It is common for genotype dependence to occur during plant somatic embryogenesis. This problem not only impedes the development of conifer reproduction on an industrial scale, but also further delays the implementation of the main tree species breeding programs. Therefore, the widespread application of somatic embryogenesis technology depends on our comprehension of the conditions that regulate the embryogenesis process and how to efficiently screen cell lines with high SE yield.

The process of conifer somatic embryogenesis consists of four stages: embryonic tissue induction, proliferation, maturation, and plant regeneration. In previous studies, early SEs (ESEs) in the embryonic tissue were observed microscopically to determine whether this embryonic tissue could be used for somatic embryogenesis. Studies have found that embryonic tissue with a large number of prominent immature SEs on the periphery has a high somatic embryogenesis potential, whereas embryonic tissue with a smooth outer periphery and almost no ESEs failed to produce SEs [10]. Furthermore, some research has analyzed calli with various somatic embryogenesis capacities and discovered that embryogenic calli with high somatic embryogenesis capacities have prominent nuclei, a high nucleus-to-cytoplasm ratio, small vacuoles, and other morphological characteristics [11,12]. Many studies, however, have demonstrated that the presence of ESEs in the embryonic tissue does not guarantee that the embryonic tissue may be utilized for somatic embryogenesis [13,14]. To further establish additional markers for selecting cell lines with high somatic embryogenesis potential, numerous studies on different plant species have been conducted. For example, in studies of *P. nigra* [15] and *P. sylvestris* [16], a high spermidine (Spd) level in the callus was used as a physiological marker of SE maturation and cell proliferation. High levels of indole-3-acetic acid (IAA) and abscisic acid (ABA) were typical characteristics of *Bactris gasipaes* and *Triticum aestivum* embryogenic calli [17,18]. High levels of reactive oxygen species (ROS), ethylene, and ROS-related proteins detected in *A. angustifolia* were associated with cell lines with high somatic embryogenesis potential [7,19]. Additionally, embryonic tissue had larger concentrations of soluble sugar and protein, and a lower level of starch when compared to nonembryonic tissue [20]. These findings suggest that morphological and physiological markers can be used to screen cell lines with a somatic embryogenesis potential before SE maturation, and that the characteristics of effective markers can also clarify the physiological basis of somatic embryogenesis. Therefore, exploring the biological factors that affect somatic embryogenesis is crucial to optimize the technology of somatic embryogenesis.

Here, we observed the numbers of pro-embryo I (PE I), pro-embryo II (PE II), pro-embryo III (PE III), and ESEs in *P. koraiensis* cell lines with various somatic embryogenesis abilities. The contents of soluble sugar, soluble protein, starch, putrescine (Put), Spd, spermine (Spm), ABA, and IAA of cell lines with different somatic embryogenesis abilities were measured. Our research aimed to obtain effective morphological and physiological markers to assess the ability of somatic embryogenesis in the early stage, and it is also expected to guide the improvement of somatic embryogenesis technology in conifers.

## 2. Results

### 2.1. P. koraiensis Somatic Embryo Development

Three different developmental stages of Proembryogenic masses (PEs) and ESEs were observed at the early stage of SE development. PE I was composed of a small clump of densely cytoplasmic embryogenic cells adjacent to a single, largely vacuolated, suspensor cell (Figure 1a). PE II had similar cell aggregates but possessed more than one vacuolated suspensor cell (Figure 1b). In PE III, enlarged clumps of highly cytoplasmatic embryogenic cells appeared more compact (Figure 1c), while ESEs had a polarized structure, composed of a cylindrical, densely cytoplasmic embryo mass (EM) and suspensor region (SR) (Figure 1d).

Even though all of the cell lines obtained had PE I, PE II, PE III, and ESE, there were significant variations in cotyledonary SE yield between cell lines during the maturation process (*p* = 0.001). We tested the SE maturation ability in 11 cell lines. Among them, only five cell lines showed cotyledonary SEs after two months of cultivation on the SE maturation medium (the cotyledonary SE yield was 45–177 SEs g^−1^ FW). The remaining cell lines did not mature in response to maturation treatment. Based on the maturation results of cotyledonary SEs, we defined the 1-1, 1-19, 1-100, 8-7, and 8-18 cell lines with somatic embryogenesis as response cell lines (R cell lines), and the 1-23, 1-102, 8-11, 8-16, 8-25, and 8-30 cell lines without somatic embryogenesis as obstructive cell lines (B cell lines) (Figure 2).

### 2.2. Proliferation Differences between Embryogenic Cell Lines of P. koraiensis

During embryogenic tissue proliferation, the FW change of the embryogenic tissue was used to compare the proliferation of R and B cell lines. We found that R and B cell lines showed similar patterns during the proliferation stage (Figure 3a). R cell lines reached the peak proliferation rate on the 8th and 13th days of culture, while B cell lines reached it on the 6th and 8th days of culture (Figure 3b).

After 14 days of proliferation, the difference in FW of embryogenic calli of R cell lines ranged from 0.17 g to 0.80 g, while those in the proliferation rate ranged from 104.9% to 532.7% (Figure 4a). In B cell lines, the difference in FW of embryogenic calli varied between 0.30 g and 0.96 g, while the proliferation rate varied between 114.9% and 519.2% (Figure 4b). In addition, we discovered a significant positive correlation between the proliferation rate and the number of PE I (R = 0.528, *p* < 0.001) and PE III (R = 0.188, *p* = 0.049) (Figure 5).

### 2.3. Differences in the Number of ESEs between Different Cell Lines of P. koraiensis

The numbers of PE III (*p* = 0.011, R = 1.9, B = 1.2) and ESEs (*p* < 0.001, R = 1.7, B = 0.6) in R cell lines were significantly higher than those in B cell lines. The number of PE III in R cell lines was 1.6-fold that of B cell lines, while the number of ESEs was 3-fold that of B cell lines. There was no significant difference in the numbers of PE I and PE II between R and B cell lines (Figure 6).

The genotype also significantly impacted PE I, PEII, PEIII, and ESE numbers when comparing the R and B cell lines (*p* < 0.05). The highest numbers of PE I, PE II, PE III, and ESEs were obtained from the 8-7, 8-11, 1-1, and 1-100 cell lines (mean = 2.9, 2.2, 3.1, and 4.1, respectively; Figure 7). In addition, a significant positive correlation between the numbers of PE III and ESEs and cotyledonary SE yield was discovered (R = 0.203, *p* = 0.034 and R = 0.652, *p* = 0.001, respectively) (Figure 5).

### 2.4. Differences in Soluble Protein, Soluble Sugar, and Starch Contents between Different Cell Lines of P. koraiensis

The soluble protein content in R cell lines was significantly higher (1.1-fold) than that in B cell lines (*p* = 0.012). In comparison to R cell lines, the starch concentration in B cell lines was significantly higher (1.2-fold) (*p* < 0.001). There was no significant difference in soluble sugar content between R and B cell lines (Figure 8).

The highest soluble protein, soluble sugar, and starch contents were observed in the 1-1, 1-102, and 1-102 cell lines, respectively (mean = 0.862 mg·mg ^−1^ FW, 3.013 mg·mg ^−1^ FW, and 2.329 mg·mg ^−1^ FW, respectively; Figure 9). In addition, it was also discovered that the soluble protein content was positively correlated with cotyledonary SE yield (R = 0.212, *p* = 0.026) and PE III number (R = 0.355, *p* < 0.001). The starch content was negatively correlated with cotyledonary SE yield (R = −0.453, *p* < 0.001) and the numbers of PE III (R = −0.377, *p* < 0.001) and ESEs (R = −0.329, *p* < 0.001). The soluble sugar content was negatively correlated with the number of PE III (R = −0.236, *p* = 0.013; Figure 5).

### 2.5. Differences in Endogenous Hormone Contents between Different Cell Lines of P. koraiensis

The IAA content in B cell lines was significantly higher (1.2-fold) than that of R cell lines (*p* = 0.003). There was no significant difference in ABA content and IAA: ABA ratio between R and B cell lines (Figure 10).

The highest values were obtained from the 1-1, 1-19, and 8-25 cell lines (mean = 66.75 ng·g^−1^, 543.69 ng·g^−1^, and 0.22, respectively; Figure 11). In addition, it was also found that the IAA content was negatively correlated with cotyledonary SE yield (R = −0.256, *p* = 0.007), while the IAA: ABA ratio was positively correlated with cotyledonary SE yield (R = 0.198, *p* = 0.038; Figure 5).

### 2.6. Differences in Endogenous Polyamine Contents between Different Cell Lines of P. koraiensis

The contents of Put and Spd in B cell lines were significantly higher (1.2-fold) than those in R cell lines (*p* < 0.001). There was no significant difference in Spm content between the two cell lines. The Put: Spd in R cell lines (10.82) was significantly higher than that in B cell lines (10.00; *p* = 0.040) when the Put: Spm and Put: Spd ratios were also compared. The Put: Spm ratio in R cell lines (0.10) was significantly lower than that in B cell lines (0.12; *p* = 0.012; Figure 12).

The highest values were observed in the 8-30, 1-102, and 8-30 cell lines, respectively (mean = 277.80 ng·g^−1^ FW, 2115.96 ng·g^−1^ FW, and 24.34 ng·g^−1^ FW, respectively; Figure 13). It was also found that the Put content was negatively correlated with cotyledonary SE yield (R = −0.249, *p* = 0.009) and the PE III number (R = −0.252, *p* = 0.008), while it was positively correlated with the PE I number (R = 0.375, *p* < 0.001). The Spm content was negatively correlated with cotyledonary SE yield (R = −0.343, *p* < 0.001) and the numbers of PE III (R = −0.253, *p* = 0.008) and ESEs (R = −0.440, *p* < 0.001). The Spd content was positively correlated with the number of ESEs (R = 0.289, *p* = 0.002; Figure 5).

The highest Put: Spd and Put: Spm were obtained in the 8-7 and 8-30 cell lines (mean = 14.38 and 0.19, respectively; Figure 14). In addition, the Put: Spd ratio was positively correlated with the number of PE I (R = 0.390, *p* < 0.001), while it was negatively correlated with the number of ESEs (R = −0.286, *p* = 0.002). The Put: Spm ratio was positively correlated with the numbers of PE I (R = 0.278, *p* = 0.003) and ESEs (R = 0.242, *p* = 0.011; Figure 5).

### 2.7. Indicator Verification

Based on the aforementioned results, we discovered that PE III and ESE numbers, soluble protein content, and Put: Spd ratio in R cell lines were significantly higher than those in B cell lines, whereas the starch, IAA, Put, and Spd contents and Put: Spm ratio in R cell lines were significantly lower than those in B cell lines. In addition, we analyzed the correlation between each indicator and cotyledonary SE yield and found that PE III and ESE numbers and soluble protein content were significantly positively correlated with SE yield, whereas the starch, ABA, Put, Spm, and Spd contents were significantly negatively correlated with cotyledonary SE yield. Therefore, we infer that PE III and ESE, soluble protein, starch, IAA, ABA, Put, Spd and Spm contents, and Put: Spd and Put: Spm ratios are closed related to SE yield.

To further confirm the relationship between these morphological and physiological indicators and SE yield, we employed some genetically unrelated cell lines (2-2, 3-5, M5-4, M13-4, M28-11, Z3, Z4, Z20, and Z21). By measuring the numbers of PE III and ESEs and the contents of soluble protein, starch, IAA, ABA, Put, Spm, and Spd (Table 1) of these cell lines, as well as analyzing the correlation between each indicator and cotyledonary SE yield, we found that the cotyledonary SE yield was significantly positively correlated with PE III (R = 0.402, *p* < 0.001) and ESE numbers (R = 0.584, *p* < 0.001), as well as Spm (R = 0.620, *p* < 0.001) and Spd contents (R = 0.301, *p* = 0.004). However, it was significantly negatively correlated with starch (R = −0.408, *p* < 0.001), ABA (R = −0.408, *p* < 0.001), and IAA contents (R = −0.504, *p* < 0.001), as well as the Put:Spd (R = −0.295, *p* = 0.005) and Put:Spm ratios (R = −0.486, *p* < 0.001; Figure 15).

According to two combined results of the two experiments, high levels of PE III and ESE can be used as morphological indicators, while low levels of starch, IAA, ABA, and Put: Spm were used as physiological indicators to identify cell lines with high SE maturation potential.

## 3. Discussion

Somatic embryogenesis in conifers typically exhibits significant genotype dependence. In the present study, 20 cell lines were tested for their ability to produce mature SEs, and only ten responded to the maturation medium. The SE yield ranged from 3 to 177 SEs g^−1^ FW. *A. angustifolia* [19] and *P. radiata* [21] both displayed a similar phenomenon, indicating that the genotype plays a significant role in determining the SE yield during somatic embryogenesis. Although somatic embryogenesis technology has been widely used in plants, SEs can only be obtained in a few genotypes. To obtain cell lines with high somatic embryogenesis potential, numerous genotypes must be tested [22]. Therefore, it is of great significance to screen embryogenic tissue that can produce a large number of SEs. According to this study, PE III and ESE counts were shown to be significantly higher in R cell lines than in B cell lines, and to be significantly positively correlated with SE yield. In conifers, embryogenic calli are under the control of plant growth regulators at different stages of PEs (PE I, PE II, and PE III) in a proliferation cycle. Among them, PE I and PE II usually died in response to ABA, while PE III was transdifferentiated into SEs after responding to ABA [23,24]. A study on Araucaria showed that the supplementation of polyamines to the culture medium can promote the development of PEs from PE II to PE III, thereby increasing the yield of SEs [25]. In addition, the addition of fluridone or the use of polyethylene glycol during pre-maturation can increase the number of PE III and ESEs, thereby increasing SE yield [26,27]. These findings indicate that the shift from PEs to ESEs is crucial for conifer somatic embryogenesis, and many embryogenic cell lines are unable to form SEs, which is largely related to the interference or blockage of the transition from PEs to ESEs [25]. The process of somatic embryogenesis is often accompanied by the accumulation and consumption of storage materials. Starch, as the main storage material in plant cells, participates in various physiological and biochemical processes in the cell, and actively participates in metabolism during the development of SEs [28]. In the studies of *Pyrus community* [20] and *Pseudotsuga menziesii* [29], when comparing embryonic tissue and nonembryonic tissue, a higher level of starch was found in nonembryonic tissue. This phenomenon is similar to the results of the present study, in which a significant negative correlation was observed between SE yield and starch of *P. koraiensis*. Similar results were also found in *P. nigra* research, which revealed that with the long-term subculture of embryonic tissue, the potential for somatic embryogenesis gradually decreased while the starch content gradually increased [30]. This suggests that high levels of starch content may negatively impact the process of somatic embryogenesis. In addition, we found that the soluble protein content in R cell lines was significantly higher than that in B cell lines. Many studies have pointed out that the protein content in embryonic tissue is positively correlated with SE yield [20,29,30]. Similarly, in the studies of *Coffea arabica* [31] and *Cordyline australis* [32], the protein content of the embryonic tissue was higher than that of the nonembryonic tissue.

In the process of embryonic tissue proliferation, there were no apparent morphological differences between *P. koraiensis* R and B cell lines, indicating that the factor affecting the SE maturation potential of *P. koraiensis* may be related to its endogenous active ingredients. In this study, the level of IAA in R cell lines was significantly lower than that in B cell lines, and there was a negative correlation between IAA content and SE yield (R = −0.256, *p* = 0.007). In *P. morrisonicola*, it was also found that the cell lines with lower somatic embryogenesis potential had a 7.5-fold higher IAA content compared with the cell lines with high somatic embryogenesis potential. In addition, studies have reported that reducing the endogenous IAA content of embryonic tissue in the process of proliferation can improve the somatic embryogenesis of cell lines with poor somatic embryogenesis [33]. The addition of auxin inhibitors in *P. sylvestris* can reduce the proliferation of PEs and significantly increase the yield of SEs, suggesting that a higher endogenous IAA has an inhibitory effect on somatic embryogenesis [34]. In addition, we found that the endogenous ABA level in embryonic tissue was significantly negatively correlated with SE yield. Studies have shown that endogenous ABA levels in embryonic tissue are harmful to the development of PE III in ESEs [35]. This view was also verified in the study of *P. abies*. Adding ABA in the early stage of somatic embryogenesis delayed the development of PE III [36]. In the *Abies cephalonica* study, it was also found that the addition of fulvic acid reduced the levels of IAA and ABA while increasing the number of PE III [26]. Moreover, in *Zea mays*, it was discovered that the embryonic tissue that lost the ability for somatic embryogenesis contained higher levels of ABA, indicating that the rise in endogenous ABA level may be the cause of the loss of embryogenic potential [37]. Therefore, in the process of SE development, PEs should not be cultured for SE maturation before PEs reach the developmental stage corresponding to early somatic embryogenesis [38]. PEs should be transferred to a medium without plant growth regulators for some time to promote the development of PEs before SE maturation, thereby increasing the yield of SEs [39].

Some studies have shown that the level of endogenous Put is related to cell growth, and Spd and Spm not only increase the SE yield of the embryonic tissue but also promote the development of SEs [15, 25]. In the current study, we discovered that B cell lines had higher levels of Put than R cell lines did. Similarly, *A. angustifolia* [19] and *P. nigra* [15] also showed higher Put contents in cell lines with low somatic embryogenesis potential. However, the verification experiment showed that the Put content in *P. koraiensis* embryonic tissue was significantly affected by genotype but was not significantly correlated with SE yield. Similarly, the addition of exogenous Put had no positive effect on SE development during proliferation or maturation in *P. abies* [40]. This demonstrates that the endogenous Put content has no significant promoting effect in the process of somatic embryogenesis. The Spd content in R cell lines of *P. koraiensis* was significantly lower than that in B cell lines. In addition, some studies have revealed that with the long-term subculture of embryogenic calli, the potential for somatic embryogenesis is diminished, and it is easy to accumulate high levels of Put and Spd in embryogenic calli [11]. However, it was found in the verification experiment that there was a positive correlation between Spd content and SE yield, and the Spd content in embryogenic calli was significantly affected by genotype. This indicates that Spd content cannot be used as an identification index for cell lines capable of somatic embryogenesis. In addition, we also found that the Put: Spm ratio in the R cell line was significantly lower than that in the B cell line, and the Put: Spm ratio was significantly negatively correlated with SE yield. These findings suggest a strong link between Put: Spm ratio and somatic embryo yield. In conclusion, we found that despite having equal SE yields, different cell lines had significant physiological differences. These physiological parameters were mostly influenced by genotype, and a similar finding was also made in research on *P. nigra* [41]. Therefore, although we obtained several physiological indicators to assess the somatic embryogenic capacity of the embryogenic callus before SE maturation culture, the effects of these traits were far less important than genotype.

## 4. Materials and Methods

### 4.1. Embryogenic Tissue Induction

Immature zygotic embryos were used to initiate embryonic tissue in *P. koraiensis*. The embryonic tissue induction medium uses mLV [42] as the basal medium, containing 27.14 μmol·L^−1^ of 2,4-dichlorophenoxyacetic acid (2,4-D), 8.88 μmol·L^−1^ of 6-benzyladenine (6-BA), 30 g·L^−1^ of sucrose, 0.5 g·L^−1^ of casein hydrolysate (CH), 0.5 g·L^−1^ of L-glutamine, and 4 g·L^−1^ of gellan gum (Phytagel, Sigma-Aldrich). To promote embryonic tissue proliferation, we transferred the obtained embryonic cell lines (1-1, 1-19, 1-23, 1-100, 1-102, 2-2, 3-5, 8-7, 8-11, 8-16, 8-18, 8-25, 8-30, M5-4, M13-4, M28-11, Z3, Z4, Z20, and Z21, of which the 1-1, 1-19, 1-23, 1-100, and 1-102 cell lines were derived from mother tree No. 1; the 8-7, 8-11, 8-16, 8-25, 8-30, and 8-18 cell lines were derived from mother tree No. 8; the 2-2, 3-5, M5-4, M13-4, M28-11, Z3, Z4, Z20, and Z21 cell lines were derived from different mother trees) to proliferation medium for propagation (mLV was used as the basic medium, supplemented with 4.52 μmol·L^−1^ of 2,4-D, 2.2 μmol·L^−1^ of 6-BA, 30 g·L^−1^ of sucrose, 0.5 g·L^−1^ of CH, and 0.5 g·L^−1^ of L-glutamine). The obtained embryonic tissue was subcultured at intervals of 2 weeks to maintain and proliferate. In this study, the embryogenic callus was used for subsequent experiments after 6 subcultures.

### 4.2. Maturation of Somatic Embryos

In the next phase of our experiment, we collected 0.35 g of embryonic tissue from the proliferation medium and resuspended it in 25 mL of liquid mLV medium (a medium without PGR and L-glutamine). The suspended tissue was then collected by pouring 5 mL of aliquots onto five filter paper discs placed in a Büchner funnel. A vacuum pulse was applied to drain the fluid, and the filter paper with cells attached was placed in maturation medium. The maturation medium contained, supplemented with 80 μM·L^−1^ of ABA, 68 g·L^−1^ of sucrose, 0.5 g·L^−1^ of CH, 0.5 L^−1^ of L-glutamine, and 12 g·L^−1^ of gellan gum. The cell lines were grown in darkness at 23 ± 1 °C for 10–12 weeks. Subculture was not required during SE maturation.

### 4.3. Screening of Morphological and Physiological Markers of High-Yield Cell Lines of Somatic Embryos

#### 4.3.1. Microscopic Observation

The 1-1, 1-19, 1-23, 1-100, 1-102, 8-7, 8-11, 8-16, 8-25, 8-30, and 8-18 cell lines were selected. After 14 days of subculture, 30 mg of the culture was randomly taken and stained with 2% acetocarmine, and the ESE morphology was observed under a Zeiss Ax microscope (Carl Zeiss, Jena, Germany). The numbers of PE I, PE II, PE III, and ESEs were recorded under a 10× magnification. This procedure was repeated 10 times and the number of ESEs was counted within 20 fields of view each time.

#### 4.3.2. Fresh Weight and Proliferation Efficiency

A total of 0.15 g of embryonic tissue was taken and transferred to fresh culture medium. The change in fresh weight (FW) was monitored every day, and the proliferation rate was calculated according to the formula below. Each measurement was repeated six times.
$$\mathrm{Proliferation}\;\mathrm{rate}\;\left(\%\right)\;\frac{\mathrm{FW}\;\mathrm{of}\;\mathrm{embryogenic}\;\mathrm{tissue}\;\mathrm{after}\;\mathrm{culture}-\mathrm{FW}\;\mathrm{of}\;\mathrm{embryogenic}\;\mathrm{tissue}\;\mathrm{before}\;\mathrm{subculture}}{\mathrm{FW}\;\mathrm{of}\;\mathrm{embryogenic}\;\mathrm{tissue}\;\mathrm{before}\;\mathrm{subculture}}\times 100\%$$

#### 4.3.3. Physiological Index Determination

After 14 days of proliferation, embryogenic tissue was collected for physiological index determination. The contents of soluble sugar, soluble protein, and starch were determined according to the method described by Peng et al. [43]. The measurement of IAA and ABA contents referred to the method described by Li et al. [44]. Approximately 0.1 g of FW of the sample was homogenized in liquid nitrogen and extracted in cold 80% (v/v) methanol with butylated hydroxytoluene (1 mmol·L^−1^) overnight at 4 °C. The supernatant was collected after centrifugation at 1000× *g* (4 °C) for 10 min, passed through a C18 Sep-Pak cartridge, and dried with nitrogen. The residue was dissolved in phosphate-buffered saline (0.01 mol·L^−1^, pH 7.4) and the content of IAA and ABA was determined. According to Dutra et al. [45], the polyamine (Put, Spm, and Spd) analysis was carried out after 14 days of embryogenic tissue subculture. Approximately 0.2–0.3 g FW of the sample was extracted in 5% (w/v) HClO_4_, and 10 mL of the culture filtrate was freeze-dried and dissolved in 5% (w/v) HClO_4_ to analyze free and soluble conjugated polyamine from the supernatant of the centrifuged extracts. The polyamine dansyl derivatives were analyzed by high-performance liquid chromatography (HPLC). Each measurement was repeated three times.

#### 4.3.4. Morphophysiological Marker Verification

To further validate the morphophysiological indicators that are closely related to SE yield in experiment 2.3, we used some genetically unrelated cell lines (2-2, 3-5, M5-4, M13-4, M28-11, Z3, Z4, Z20, and Z21) to repeat the experiments on these morphological and physiological indicators.

#### 4.3.5. Statistical Analysis

Data analysis was performed using Microsoft Excel 2019. SPSS (v21.0, SPSS Inc.) software was used for the one-way analysis of variance (ANOVA), Duncan’s multiple comparisons (Duncan, α = 0.05), and Pearson bivariate correlation analysis. SigmaPlot (v12.5, SYSTAT) as well as Microsoft Visio 2016 software was used for graph generation.

## 5. Conclusions

In conclusion, this study compares the morphological and physiological characteristics of cell lines with various somatic embryogenesis capabilities. We discovered that the numbers of PE III and ESEs, the soluble protein content, and the Put: Spd ratio in R cell lines were significantly higher than those in B cell lines, which were 1.6-, 3-, and 1.1-fold those of B cell lines, respectively. The starch, IAA, Put, and Spd contents and the Put: Spm ratio in B cell lines were significantly higher than those in R cell lines, which were 1.2-fold those of R cell lines. In addition, the PE III and ESE numbers and soluble protein content were significantly positively correlated with SE yield, while the starch, ABA, Put, Spm, and Spd contents were significantly negatively correlated with SE yield. Through further verification of the above indicators, it was found that the numbers of PE III and ESE were significantly positively correlated with SE yield, whereas the quantities of starch, IAA, and ABA, and the Put: Spm ratio were significantly negatively correlated with SE yield. Therefore, we recommend using high levels of PE III and ESEs as morphological indicators, and low levels of starch, IAA, ABA, and Put: Spm as physiological markers for screening cell lines with a high somatic embryogenesis potential at an early stage. In addition, we also found that the relationship between Spd and Spm and somatic embryo yield was opposite in the two experimental results. Therefore, we speculate that the content differences of Spd and Spm are mainly affected by genotype. In general, our results provide a new perspective for the early screening of cell lines with high SE maturation potential and provide a theoretical basis for subsequent studies based on differences in somatic embryogenesis. Additionally, the found markers were used as indicators to develop the treatments along the SE propagation to find ones that result in the majority of the genotypes showing indicators favorable for high embryo production. As a result, the widespread application of somatic embryogenesis in forest tree breeding was promoted.

## Figures and Tables

**Figure 1 plants-11-01867-f001:**
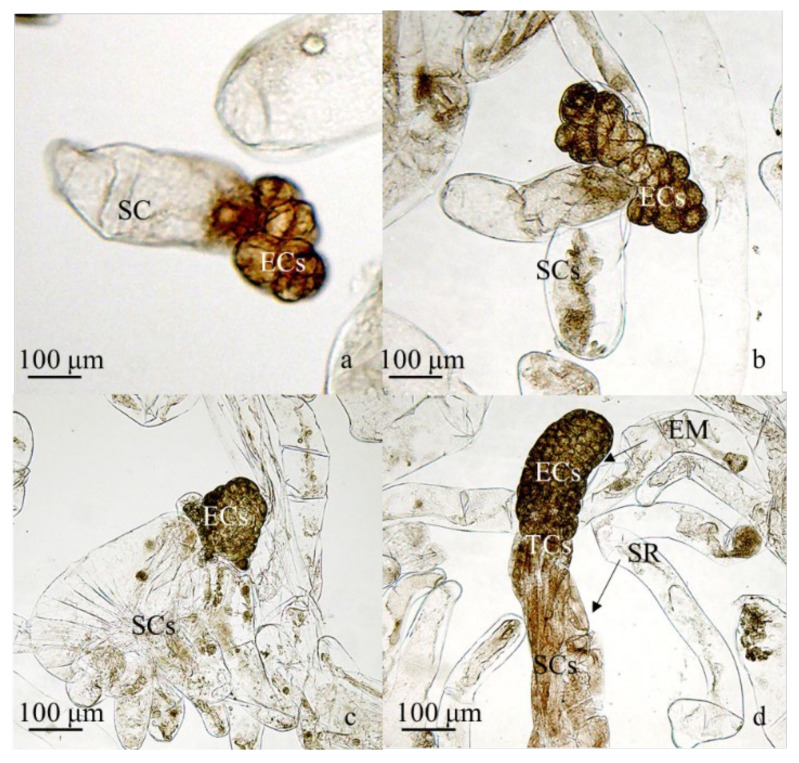
Characterization of the *P. koraiensis* embryogenic tissue. (**a**) PE I, bar = 100 μm; (**b**) PE II, bar = 100 μm; (**c**) PE III, bar = 100 μm; (**d**) ESE, bar = 100 μm. EM: embryo mass; TC: embryonic tube cells; SR: suspensor region; EC: embryonic cell; SC: suspensor cord-like cell.

**Figure 2 plants-11-01867-f002:**
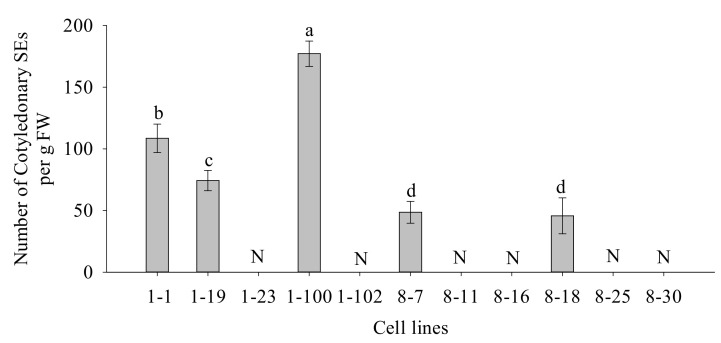
Number of cotyledonary SEs in different cell lines of *P. koraiensis*. Different letters above the columns indicate significant differences at the *p* < 0.05 level, and “N” indicates that the number of cotyledonary SEs was 0.

**Figure 3 plants-11-01867-f003:**
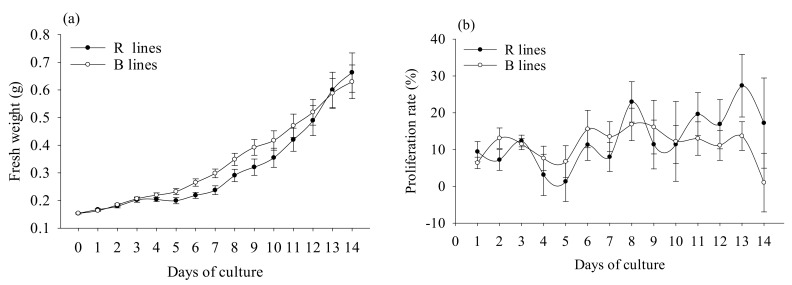
Changes in *P. koraiensis* R and B cell lines during proliferation. (**a**) Fresh weight (FW) changes in R and B cell lines during proliferation. (**b**) Changes in proliferation rate in R and B cell lines during proliferation.

**Figure 4 plants-11-01867-f004:**
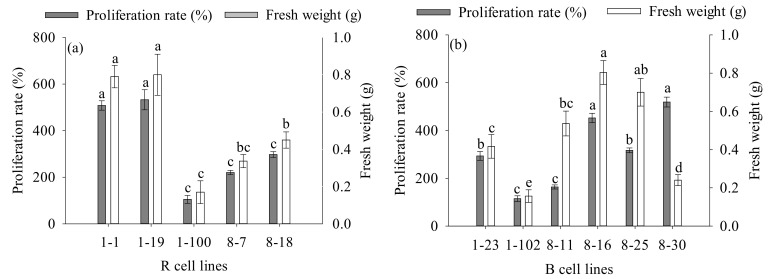
Differences in FW and proliferation rate of different cell lines of *P. koraiensis* after 14 days of proliferation. (**a**) R cell lines; (**b**) B cell lines. Different letters above the column indicate significant differences at the *p* < 0.05 level.

**Figure 5 plants-11-01867-f005:**
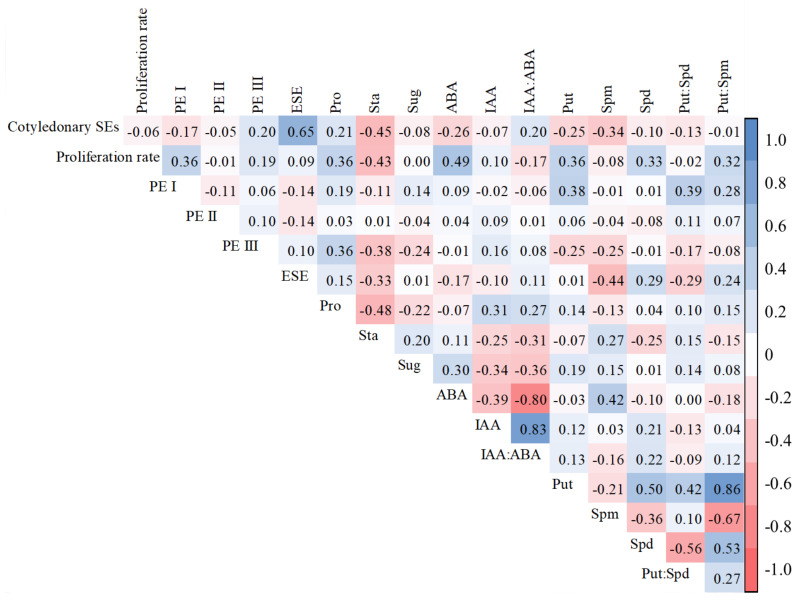
Correlation between cotyledonary SE yield and morphological and physiological indicators in different cell lines of *P. koraiensis.* The numbers in the figure are correlation coefficients (R-values). The larger the R-value, the stronger the correlation.

**Figure 6 plants-11-01867-f006:**
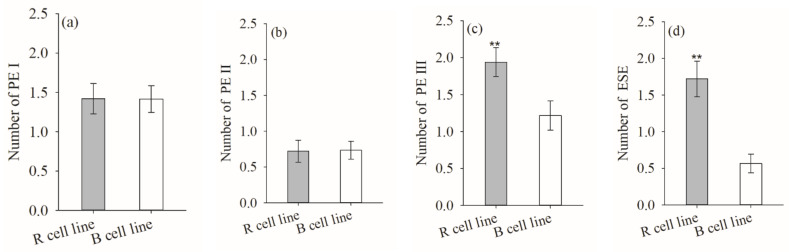
Differences in the numbers of PE I, PE II, PE III, and ESEs between *P. koraiensis* R and B cell lines. ** indicates that the difference is significant at the *p* < 0.01 level.

**Figure 7 plants-11-01867-f007:**
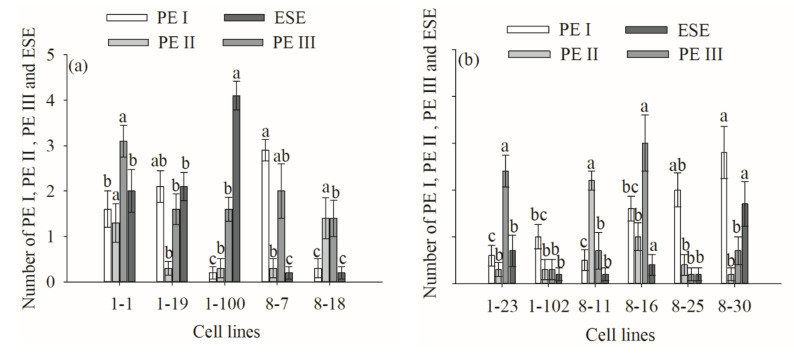
Differences in the number of ESEs between different cell lines of *P. koraiensis*. (**a**) R cell lines; (**b**) B cell lines. Different letters above the column indicate significant differences at the *p* < 0.05 level.

**Figure 8 plants-11-01867-f008:**
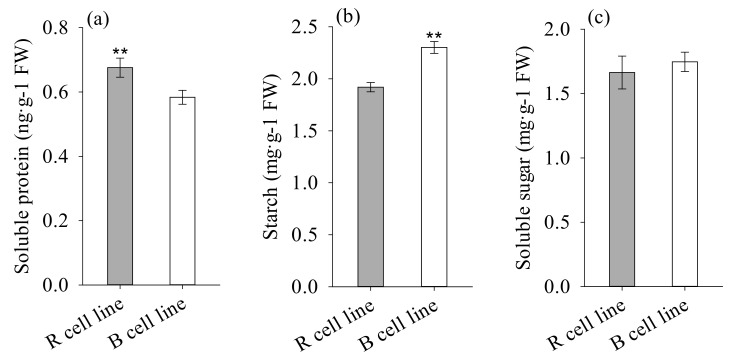
Differences in soluble protein, starch, and soluble sugar contents between *P. koraiensis* R and B cell lines. ** above the column indicates that the difference is significant at the *p* < 0.01 level.

**Figure 9 plants-11-01867-f009:**
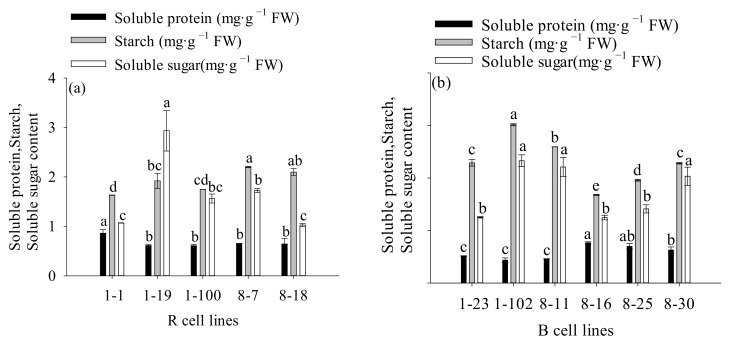
Differences in soluble sugar, starch, and soluble protein contents between different cell lines of *P. koraiensis*. (**a**) R cell lines; (**b**) B cell lines. Different letters above the bar indicate significant differences at the *p* < 0.05 level.

**Figure 10 plants-11-01867-f010:**
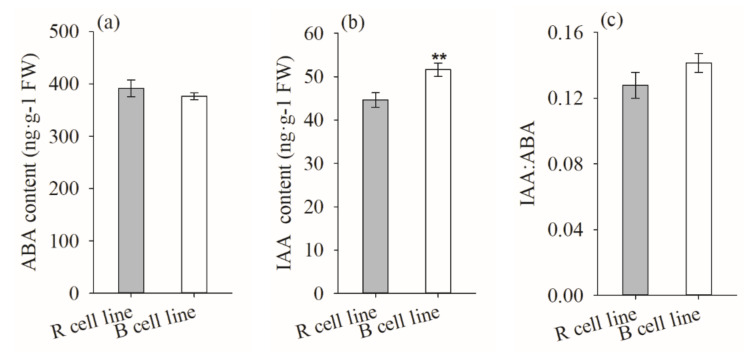
Differences in IAA and ABA contents and the IAA: ABA ratio between *P. koraiensis* R and B cell lines. ** above the column indicates that the difference is significant at the *p* < 0.01 level.

**Figure 11 plants-11-01867-f011:**
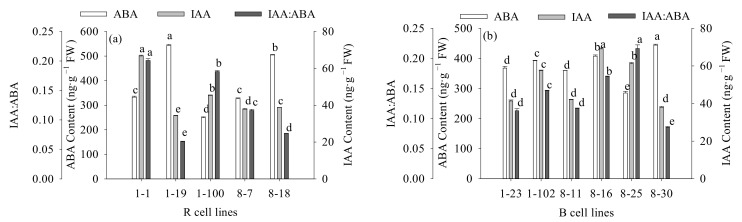
Differences in ABA and IAA contents between different cell lines of *P. koraiensis*. (**a**) R cell lines; (**b**) B cell lines. Different letters above the columns indicate significant differences at the *p* < 0.05 level.

**Figure 12 plants-11-01867-f012:**
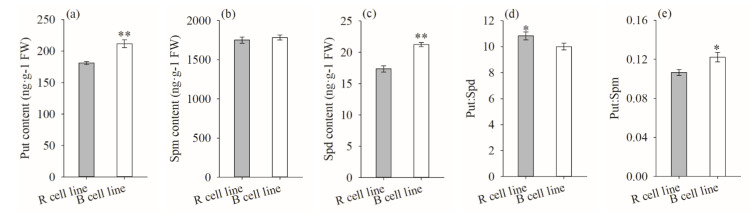
Differences in Put, Spm, and Spd contents, as well as the Put: Spd and Put: Spm ratios between *P. koraiensis* R and B cell lines. ** above the column indicates that the difference is significant at the *p* < 0.01 level, and * above the column indicates that the difference is significant at the *p* < 0.05 level.

**Figure 13 plants-11-01867-f013:**
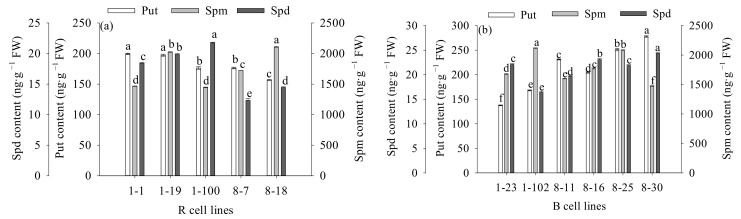
Differences in Put, Spd, and Spm contents between different cell lines of *P. koraiensis*. (**a**) R cell lines; (**b**) B cell lines. Different letters above the columns indicate significant differences at the *p* < 0.05 level.

**Figure 14 plants-11-01867-f014:**
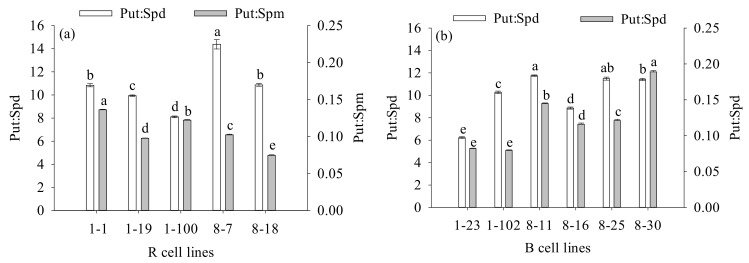
Differences in the Put: Spd and Put: Spm ratios between different cell lines of *P. koraiensis.* (**a**) R cell lines; (**b**) B cell lines. Different letters above the column indicate significant differences at the *p* < 0.05 level.

**Figure 15 plants-11-01867-f015:**
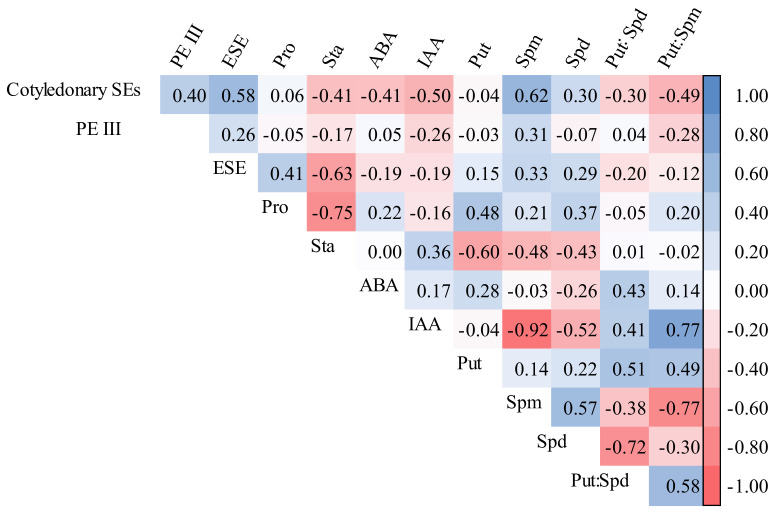
Correlations between SE yield and morphological and physiological indicators of *P. koraiensis* cell lines. The numbers in the figure are correlation coefficients (R-values); the larger the R-value, the more significant the correlation.

**Table 1 plants-11-01867-t001:** Cotyledonary SE yield, PE III, ESE, soluble protein, starch, ABA, IAA, Put, Spm, and Spd content, and Put: Spd and Put: Spm differences between different cell lines of *P. koraiensis*.

Cell line	Cotyledonary SEs	PEMIII	ESE	Pro	Sta	ABA
	SE·g ^−1^ FW	Number	Number	ng·g ^−1^ FW	ng·g ^−1^ FW	ng·g ^−1^ FW
2-2	2.86 ± 1.90	2.00 ± 0.67	2.60 ± 0.43	0.78 ± 0.08	1.59 ± 0.08	409.84 ± 1.52
3-5	5.71 ± 2.33	1.80 ± 0.33	2.80 ± 0.33	1.08 ± 0.03	1.32 ± 0.01	413.28 ± 3.51
m5-4	0 ± 0	0.50 ± 0.31	1.00 ± 0.21	0.60 ± 0.01	2.18 ± 0.00	293.36 ± 2.82
m13-4	25.71 ± 7.62	1.70 ± 0.65	0.90 ± 0.48	0.91 ± 0.05	1.59 ± 0.02	438.78 ± 5.11
m28-11	0 ± 0	0.30 ± 0.21	0.10 ± 0.10	0.35 ± 0.02	4.03 ± 0.01	276.72 ± 3.78
z3	0 ± 0	2.60 ± 0.37	0.10 ± 0.10	0.30 ± 0.00	4.43 ± 0.02	441.68 ± 1.18
z4	0 ± 0	2.40 ± 0.48	0.20 ± 0.13	0.48 ± 0.03	3.03 ± 0.03	353.77 ± 1.96
z20	0 ± 0	1.90 ± 0.43	0.80 ± 0.29	0.55 ± 0.00	2.48 ± 0.03	512.76 ± 0.72
z21	157.14 ± 19.05	3.80 ± 0.65	4.40 ± 0.62	0.61 ± 0.02	1.23 ± 0.01	264.78 ± 3.38
Cell line	IAA	Put	Spm	Spd	Put:Spd	Put:Spm
	ng·g ^−1^ FW	ng·g ^−1^ FW	ng·g ^−1^ FW	ng·g ^−1^ FW		
2-2	56.32 ± 0.54	232.01 ± 1.68	1761.09 ± 8.33	23.31 ± 0.16	9.96 ± 0.12	0.132 ± 0.001
3-5	69.43 ± 0.19	221.41 ± 1.03	1153.37 ± 24.49	17.25 ± 0.19	12.85 ± 0.19	0.193 ± 0.004
m5-4	59.71 ± 0.19	193.57 ± 0.90	1385.82 ± 8.49	16.27 ± 0.11	11.90 ± 0.05	0.140 ± 0.001
m13-4	37.83 ± 0.39	228.61 ± 2.17	2376.09 ± 9.52	25.52 ± 0.15	8.96 ± 0.05	0.096 ± 0.001
m28-11	67.09 ± 0.1	192.70 ± 2.41	1201.47 ± 5.95	22.40 ± 0.24	8.62 ± 0.19	0.160 ± 0.002
z3	59.83 ± 0.42	135.12 ± 2.03	1484.19 ± 15.58	13.20 ± 0.10	10.24 ± 0.15	0.091 ± 0.002
z4	53.79 ± 0.38	230.11 ± 0.63	1491.48 ± 16.14	14.00 ± 0.18	16.46 ± 0.17	0.154 ± 0.002
z20	68.91 ± 0.45	251.78 ± 2.28	1462.33 ± 11.07	14.02 ± 0.08	17.96 ± 0.22	0.172 ± 0.002
z21	44.71 ± 0.26	201.16 ± 1.29	2315.61 ± 12.07	21.42 ± 0.06	9.39 ± 0.04	0.087 ± 0.001

## Data Availability

Not applicable.

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
