# Peer review of "Morphological and Physiological Indicators for Screening Cell Lines with High Potential for Somatic Embryo Maturation at an Early Stage of Somatic Embryogenesis in Pinus Koraiensis"

_plants, 2022, doi:10.3390/plants11141867_

Round 1

Reviewer 1 Report

The topic of the ms is of great interest, i.e. it would indeed be beneficial to be able to predict prior to maturation which cell lines are able to produce (high numbers) of mature somatic embryos. Unfortunately the current ms does not give any novel answers to this, and due to loosely written M & M, it is difficult to say if the results presented are sound.

The ms underlines correlations between SE yield and several studied factors. One of the major issues is that it is, however, not clear if these correlations were correctly calculated, as it is not described in statistical analyses but only ANOVA is mentioned. One should note that the SE data is not normally distributed, and the number of cell lines is small - and if this was taken into account,  is not told. 

The authors present results from both original and verification experiments, and the results differ to some extent among these, when looking at the emphasized correlations. The authors, however, forget this e.g. in the case of Spm correlations that are discussed in r 344-353 reporting only the ones fitting to their discussion...

Other shortcomings in the M & M:

- pollen parents not told for controlled creossings

-description of maturation method is totally missing, i.e. how (as tissue clumps or on filter, if material was weighed, replications etc..) and how it was taken care that the treatment was comparablefor every line (knowing that e.g. tissue clump size or weight/filter affects SE production)

- not told if marker candidates were studied using tissue from proliferation medium, assuming so ?

- description of proliferation efficiency testing is not clear- based on the text one could  the authors subcultured tissue every day, but hardly so ?

- not told when samples for IAA and ABA, sugar, starch and proteins were taken

- not told how verification was done, by repeating almost all experiments but not quite all of them ??

It is also amazing if the authors were really able to obtain enough material for both maturations and for all the other experiments following only two subcultures on proliferation medium, as the text says... I found it hard to believe. Or maybe only maturations were done then.. but then it should be told when (i.e. after how many subcultures) other experiments were performed.

The authors call embryogenic cell lines throughout the ms as "callus" or calli". This is wrong as callus means undifferentiated tissue - "callus/calli" should be replaced e.g. by "tissue" or "culture".

There is a lot of repetition of the results by both text and figures. There is e.g.  no need to tell genotypic differences in  detail in the text when these are clearly seen in the figures. By the way - for genotypic differences, it is not told if the comparisons were made over both R and B lines, or separately for these two groups.  In the figures, the letters indicating significant differences don't always make sense i.e. needs to be checked, but hard to say what is correct when not knowing which lines were included in the comparison.

There are many other points that could be improved too, e.g. Introduction gives too positive picture on practical applications of SE... potentials are high but not yet reality.  Discussion does not touch the point if the production of cotyldonary embryos quarentee plant regeneration. Well, it does not. It would be good to know if the produced SE were able to germinate and convert into plants too ? Further, I am also missing a critical evaluation if the found markers really provide a way to select cell lines e.g. for mass-propagation prior to maturation ?? Or if the safest way is still to just test their SE production capacity ??

Overall, the ms is somehow carelessly written. For example, discussion starts with instructions to the authors and Supplementary data refers to Tree Physiology. Maybe this ms was originally submitted there and rejected ? 

Author Response

Response to Reviewer 1 Comments

Dear Reviewer,

Our sincere thanks to you for the time and effort that you have put into reviewing our manuscript! We found all the comments very constructive and helpful, and have revised our manuscript according to all comments. Please find, below, our point-by-point response to the comments raised.

Thank you for considering our revised manuscript!

Point 1: The topic of the ms is of great interest, i.e. it would indeed be beneficial to be able to predict prior to maturation which cell lines are able to produce (high numbers) of mature somatic embryos. Unfortunately the current ms does not give any novel answers to this, and due to loosely written M & M, it is difficult to say if the results presented are sound.

Response 1: This article is an initial exploration of which cell lines are predicted to produce (massive) mature somatic embryos prior to maturation, and in subsequent studies, we will conduct extensive experiments to verify the validity of the results.

Point 2: The ms underlines correlations between SE yield and several studied factors. One of the major issues is that it is, however, not clear if these correlations were correctly calculated, as it is not described in statistical analyses but only ANOVA is mentioned. One should note that the SE data is not normally distributed, and the number of cell lines is small - and if this was taken into account,  is not told. 

Response 2: Thank you reviewers for your comments on this article. Because there are few cell lines obtained in our laboratory at present, this article is only a preliminary exploration of the factors affecting SE production, and we will use a large number of cell lines for verification in the follow-up study.

Point 3: The authors present results from both original and verification experiments, and the results differ to some extent among these, when looking at the emphasized correlations. The authors, however, forget this e.g. in the case of Spm correlations that are discussed in r 344-353 reporting only the ones fitting to their discussion...

Response 3: We have revised for the discussion between Spm and somatic embryo yield.

Point 4: - pollen parents not told for controlled creossings

Response 4: It is not known whether pollen parents control pollination because of missing data from forest farms

Point 5: -description of maturation method is totally missing, i.e. how (as tissue clumps or on filter, if material was weighed, replications etc..) and how it was taken care that the treatment was comparablefor every line (knowing that e.g. tissue clump size or weight/filter affects SE production)

Response 5: Complete somatic embryo maturation methods are supplemented in the text

Point 6: - not told if marker candidates were studied using tissue from proliferation medium, assuming so ?

Response 6: In a second validation experiment, we investigated candidate markers from the first laboratory using tissue in proliferation medium

Point 7: - description of proliferation efficiency testing is not clear- based on the text one could  the authors subcultured tissue every day, but hardly so ?

Response 7: No subculture treatment was performed during this period.

Point 8: - not told when samples for IAA and ABA, sugar, starch and proteins were taken

Response 8: The specific time has been added in the original text " After 14 days of proliferation, embryogenic callus was collected for physiological index determination."

Point 9: - not told how verification was done, by repeating almost all experiments but not quite all of them ??

Response 9: Validation experiments are landmark indicators obtained by repeating experiment 4.3.

Point 10: It is also amazing if the authors were really able to obtain enough material for both maturations and for all the other experiments following only two subcultures on proliferation medium, as the text says... I found it hard to believe. Or maybe only maturations were done then.. but then it should be told when (i.e. after how many subcultures) other experiments were performed.

Response 10: Our experiment used three-month-old embryogenic callus as the research material, so the embryogenic callus had been subcultured for three months when this experiment was carried out. Therefore, the research material is sufficient.

Point 11: The authors call embryogenic cell lines throughout the ms as "callus" or calli". This is wrong as callus means undifferentiated tissue - "callus/calli" should be replaced e.g. by "tissue" or "culture".

Response 11: “callus” has been replaced by “tissue”

Point 12: There is a lot of repetition of the results by both text and figures. There is e.g.  no need to tell genotypic differences in  detail in the text when these are clearly seen in the figures. By the way - for genotypic differences, it is not told if the comparisons were made over both R and B lines, or separately for these two groups.  In the figures, the letters indicating significant differences don't always make sense i.e. needs to be checked, but hard to say what is correct when not knowing which lines were included in the comparison.

Response 12: We've made corrections to duplicate results for text and numbers.

Point 13: There are many other points that could be improved too, e.g. Introduction gives too positive picture on practical applications of SE... potentials are high but not yet reality.  Discussion does not touch the point if the production of cotyldonary embryos quarentee plant regeneration. Well, it does not. It would be good to know if the produced SE were able to germinate and convert into plants too ? Further, I am also missing a critical evaluation if the found markers really provide a way to select cell lines e.g. for mass-propagation prior to maturation ?? Or if the safest way is still to just test their SE production capacity ??

Response 13: We have revised the full text

Point 14: Overall, the ms is somehow carelessly written. For example, discussion starts with instructions to the authors and Supplementary data refers to Tree Physiology. Maybe this ms was originally submitted there and rejected ? 

Response 14: Done. Please see above.

Reviewer 2 Report

Line 43. Eliminate the ..

Line 46 and all the citations must be corrected. The citations are not superscripts.

Line 75 indole-3 acetic must be indole-3-acetic

Line 76. The scientific names of species must be in italics.

Line 97. Redaction. Eliminate “and vacuolated cell”.

Line 105. The “P” must be italic “P”.

Line 107. g-1 must be g-1

Line 209. Figures and tables should go as close as possible to their description, not at the end of the results.

Line 240 & 248. IAA:     ABA must be AIA:ABA in both figures.

Line 250 Put:    Spm must be Put:Spm.

Lines 260-263. This paragraph must be deleted.

Lines 368-369, 378. The concentrations of the plant growth regulators must be in molarity.

Line 369. The name benzyl amino-purine must be change to benzyladenine.

Line 386. 23±1ºC must be 23 ± 1 ºC.

Lines 408, 409. 4ºC must be 4 ºC,

Line 458. All the references must be corrected. The name of the journals must be in capitals. PCTOC must be eliminated.

This manuscript presents an exhaustive investigation of different aspects affecting somatic embryogenesis in Pinus koraiensis. A large number of experiments and measurements of different factors are described. My main concern with this manuscript is that critical factors have not been considered and must be considered to reach any real conclusions. The first is the lack of mention of the role of cytokinins in the SE process. Auxins and cytokinins interact with each other. One affects the other. So if the auxin concentration changes, the cytokinin concentration changes. The other aspect that is not mentioned is the auxin conjugates. In general, auxins are found in plant tissues, including embryogenic systems, in the form of conjugates (generally more than 95%), and free auxin is less than 5%. The authors should take this into account if they want to improve the SE process in Pinus koraiensis.

The manuscript requires a thorough revision of the English. As an example, this reviewer has edited the abstract.

Abstract

Many cell lines in the embryogenic callus cannot produce somatic embryos (SEs) even if they meet the optimal SE maturation culture conditions during conifer somatic embryogenesis. This phenomenon hinders the progress of industrial-scale reproduction of conifers. Therefore, there is an urgent need to obtain morphological and physiological markers to screen embryogenic calli in response to SE maturation conditions. To detect cell lines with high somatic embryogenesis potential during the proliferation process, we counted the number of pro-embryo and early SE (ESE) in different cell lines and storage substances, endogenous hormones, and polyamine content. The results showed that the yield of P. koraiensis SEs was heavily dependent on genotype (P = 0.001). There were high levels of PE III (pro-embryo III) number, ESE number, soluble protein content, and putrescine: spermidine (Put:Spd) in the response cell lines (R cell lines), which were 1.6-, 3-, 1.1-, and 1-fold those of the obstructive cell lines (B cell lines), respectively. The B cell line had high levels of starch, auxin (IAA), Put, Spd, and putrescine:spermine (Put: Spm) compared to the R cell line. In addition, the numbers of PE III and ESEs and soluble protein content were significantly positively correlated with SE yield.

In contrast, the contents of starch, abscisic acid (ABA), Put, Spm, and Spd were significantly negatively correlated with SE yield. We validated this result with nine cell lines. The PE III and ESE numbers and the Spm and Spd contents were positively correlated with SE yield, while the levels of starch, ABA, IAA, Put:Spd, and Put:Spm was negatively correlated with SE yield. Thus, we recommend using high PE III and ESEs as morphological indicators and low levels of starch, IAA, ABA, and Put:Spm as physiological markers to screen cell lines with high somatic embryogenesis potential.

Author Response

Response to Reviewer 2 Comments

Dear Reviewer,

Our sincere thanks to you for the time and effort that you have put into reviewing our manuscript! We found all the comments very constructive and helpful, and have revised our manuscript according to all comments. Please find, below, our point-by-point response to the comments raised.

Thank you for considering our revised manuscript!

Point 1: The ms underlines correlations between SE yield and several studied factors. One of the major issues is that it is, however, not clear if these correlations were correctly calculated, as it is not described in statistical analyses but only ANOVA is mentioned. One should note that the SE data is not normally distributed, and the number of cell lines is small - and if this was taken into account, is not told.

Response 1: Thank you reviewers for your comments on this article. Because there are few cell lines obtained in our laboratory at present, this article is only a preliminary exploration of the factors affecting SE production, and we will use a large number of cell lines for verification in the follow-up study.

Point 2: The authors present results from both original and verification experiments, and the results differ to some extent among these, when looking at the emphasized correlations. The authors, however, forget this e.g. in the case of Spm correlations that are discussed in r 344-353 reporting only the ones fitting to their discussion...

Response 2: We have revised for the discussion between Spm and somatic embryo yield.

Point 3: - pollen parents not told for controlled creossings

Response 3: It is not known whether pollen parents control pollination because of missing data from forest farms

Point 4: -description of maturation method is totally missing, i.e. how (as tissue clumps or on filter, if material was weighed, replications etc..) and how it was taken care that the treatment was comparablefor every line (knowing that e.g. tissue clump size or weight/filter affects SE production)

Response 4: Complete somatic embryo maturation methods are supplemented in the text

Point 5: - not told if marker candidates were studied using tissue from proliferation medium, assuming so ?

Response 5: In a second validation experiment, we investigated candidate markers from the first experiment using tissue in proliferation medium

Point 6: - description of proliferation efficiency testing is not clear- based on the text one could  the authors subcultured tissue every day, but hardly so ?

Response 6: No subculture treatment was performed during this period.

Point 7: - not told when samples for IAA and ABA, sugar, starch and proteins were taken

Response 7: The specific time has been added in the original text ". After 14 days of proliferation, embryogenic callus was collected for physiological index determination."

Point 8: - not told how verification was done, by repeating almost all experiments but not quite all of them ??

Response 8: Validation experiments are landmark indicators obtained by repeating experiment 4.3.

Point 9: It is also amazing if the authors were really able to obtain enough material for both maturations and for all the other experiments following only two subcultures on proliferation medium, as the text says... I found it hard to believe. Or maybe only maturations were done then. but then it should be told when (i.e. after how many subcultures) other experiments were performed.

Response 9: Our experiment used three-month-old embryogenic callus as the research material, so the embryogenic callus had been subcultured for three months when this experiment was carried out. Therefore, the research material is sufficient.

Point 10: The authors call embryogenic cell lines throughout the ms as "callus" or calli". This is wrong as callus means undifferentiated tissue - "callus/calli" should be replaced e.g. by "tissue" or "culture".

Response 10: “callus” has been replaced by “tissue”

Point 11: There is a lot of repetition of the results by both text and figures. There is e.g.  no need to tell genotypic differences in  detail in the text when these are clearly seen in the figures. By the way - for genotypic differences, it is not told if the comparisons were made over both R and B lines, or separately for these two groups.  In the figures, the letters indicating significant differences don't always make sense i.e. needs to be checked, but hard to say what is correct when not knowing which lines were included in the comparison.

Response 11: We've made corrections to duplicate results for text and numbers.

Point 12: Line 43. Eliminate the ..

Response 12: Done. Please see above.

Point 13: Line 46 and all the citations must be corrected. The citations are not superscripts.

Response 13: Done. Please see above.

Point 14: Line 75 indole-3 acetic must be indole-3-acetic

Response 14: “indole-3 acetic” has been replaced by “indole-3-acetic”

Point 15: The scientific names of species must be in italics.

Response 15: Done. Please see above.

Point 16: Redaction. Eliminate “and vacuolated cell”.

Response 16: Done. Please see above.

Point 17: The “P” must be italic “P”

Response 17: “P” has been replaced by “P”

Point 18: g-1 must be g-1

Response 18: “g-1” has been replaced by “g-1

Point 19: Figures and tables should go as close as possible to their description, not at the end of the results.

Response 19: We further supplement the description of the figures and tables

Poin 20: IAA:     ABA must be AIA:ABA in both figures.

Response 20: Done. Please see above.

Point 21: Put:    Spm must be Put:Spm

Response 21: Done. Please see above.

Point 22: This paragraph must be deleted.

Response 22: Done. Please see above.

Point 23: The concentrations of the plant growth regulators must be in molarity.

Response 23: We have changed the concentration of plant growth regulators to molarity.

Point 24: The name benzyl amino-purine must be change to benzyladenine.

Response 24: “benzyl amino-purine” has been replaced by “benzyladenine”

Point 25: 23±1ºC must be 23 ± 1 ºC.

Response 25: “23±1ºC” has been replaced by “23 ± 1 ºC”

Point 26: 4ºC must be 4 ºC,

Response 26: “4ºC” has been replaced by “4 ºC”

Point 27: All the references must be corrected. The name of the journals must be in capitals. PCTOC must be eliminated.

Response 27: We corrected citations.

Point 28: Subsection 2.1. and Legend to the Figure 1: The term ‘suspension’ is incorrect here and must be replaced with the term ‘suspensor’

Response 28: “suspension” has been replaced by “suspensor”

Point 29: Figure 5 and section Discussion: Abbreviations should be used consistently throughout the text and figures. So, it should be either PE or PEM in the entire document (and not both of them). Since the majority of figures and the main text contains PE (I, II, III), I suggest that the authors make appropriate corrections.

Response 29: Done. Please see above.

Point 30: Figure 8, 9: What is prot in (ng per g prot)? If prot is an abbreviation for ‘protein’, then why is starch/soluble sugar content expressed per g protein (and not per g FW)? If this was a mistake, and there should be FW instead, then the authors are advised to check carefully all the graphs and make corrections in both figures and the main text.

Response 30: This was a bug and we have corrected the units in the diagram and text.

Point 31: Figure 13 has been mistaken for Fig. 11, and contains WRONG GRAPHS (!!!), showing levels of IAA, ABA instead of showing levels of Put, Spm and Spd. This needs to be corrected, and the values for Put, Spd and Spm, for each cell line, should be presented instead!

Response 31: Done. Please see above.

Point 32: References [29] and [31] are identical, so that [31] should be removed and all the references following ref. #30 should be corrected accordingly

Response 32: Done. Please see above.

Point 33: There are many other points that could be improved too, e.g. Introduction gives too positive picture on practical applications of SE... potentials are high but not yet reality.  Discussion does not touch the point if the production of cotyldonary embryos quarentee plant regeneration. Well, it does not. It would be good to know if the produced SE were able to germinate and convert into plants too ? Further, I am also missing a critical evaluation if the found markers really provide a way to select cell lines e.g. for mass-propagation prior to maturation ?? Or if the safest way is still to just test their SE production capacity ??

Response 33: We have corrected the shortcomings of the full text

Reviewer 3 Report

Comments concerning manuscript plants-1751823

This research represents a continuation of an integrated study of somatic embryogenesis in Pinus koraiensis. By comparing the morphological and physiological characteristics between contrasting genotypes of Pinus koraiensis to identify and characterize cell lines with high SE maturation potential during the proliferation process, the authors have identified biomarkers associated with embryogenic ability in this conifer species. In line with findings that the early events in somatic embryogenesis are crucial for the successful completion of the overall embryogenic program, screening for the cell lines with high SE maturation potential early during the process would be a valuable tool for the future high throughput analyses that could contribute to better understanding of molecular mechanisms involved in the regulation of embryo development.

The manuscript is well written, and clearly presented. The authors provided sufficient background and included all relevant references. The research design is appropriate and the methods used are described adequately. The results are clearly presented and support the conclusions. There are, however, some minor issues, and in order to make this paper publishable, the authors need to respond to suggested corrections and comments.

Suggestions and corrections that would help improve the manuscript are included in the revised pdf version (plants-1751823-peer-review-v1-revised). Listed below are some suggested corrections and comments to the text and figures that the authors should respond to:

Title

It is apparent [from e.g. (Line 103) “Although PE I, PE II, PE III, and ESE were present in all cell lines obtained, there were significant differences in SE yield between cell lines during the maturation culture process”] that the phrase “somatic embryogenesis potential”, in this work, does not refer to SE induction, but rather to the maturation of somatic embryos. Therefore I consider that the ‘potential’ in the phrase ‘high somatic embryogenesis potential’ should be specified further, to denote the potential of (early) SEs for maturation. Furthermore, the expression used in the title [“at an early stage”] may be somewhat unclear to a broader audience, and I suggest specifying, in the very title, that it is all about screening at an early stage of somatic embryogenesis or at an early stage of somatic embryo development or, alternatively, early screening.

One example of the title might be: Morphological and physiological indicators for screening cell lines with high potential for somatic embryo maturation at an early stage of somatic embryogenesis in Pinus koraiensis

Results

Subsection 2.1. and Legend to the Figure 1: The term ‘suspension’ is incorrect here and must be replaced with the term ‘suspensor

Figure 5 and section Discussion: Abbreviations should be used consistently throughout the text and figures. So, it should be either PE or PEM in the entire document (and not both of them). Since the majority of figures and the main text contains PE (I, II, III), I suggest that the authors make appropriate corrections.

Figure 8, 9: What is prot in (ng per g prot)?

If prot is an abbreviation for ‘protein’, then why is starch/soluble sugar content expressed per g protein (and not per g FW)?

If this was a mistake, and there should be FW instead, then the authors are advised to check carefully all the graphs and make corrections in both figures and the main text.

Figure 13 has been mistaken for Fig. 11, and contains WRONG GRAPHS (!!!), showing levels of IAA, ABA instead of showing levels of Put, Spm and Spd. This needs to be corrected, and the values for Put, Spd and Spm, for each cell line, should be presented instead!

Discussion/References

References [29] and [31] are identical, so that [31] should be removed and all the references following ref. #30 should be corrected accordingly

Author Response

Response to Reviewer 3 Comments

Dear Reviewer,

Our sincere thanks to you for the time and effort that you have put into reviewing our manuscript! We found all the comments very constructive and helpful, and have revised our manuscript according to all comments. Please find, below, our point-by-point response to the comments raised.

Thank you for considering our revised manuscript!

Point 1: if X is 1-fold higher than Y (in the sense that '-fold' is used throughout the text), then X=Y; in which case it makes no sense saying that X is higher than Y, at all. According to the Fig. 12, this Put:Spd ratio appears to be 1.1

Response 1: I have removed the Put:Spd ratio in the Abstract.

Point 2: the term suspension, which is used here and also in the legend for Figure 1 must be avoided, and changed into 'suspensor' here and elsewhere in the text

Response 2: “suspension” has been replaced by “suspensor”

Point 3: THIS IS WHY THE TITLE SHOULD BE CHANGED

Response 3: “Morphological and physiological indicators for screening cell lines with high somatic embryogenesis potential at an early 3stage in Pinus koraiensis” has been replaced by “Morphological and physiological indicators for screening cell lines with high potential for somatic embryo maturation at an early stage of somatic embryogenesis in Pinus koraiensis”

Point 4: Alternatively, it can be said SEs of the remaining cell lines did not mature in response to [maturation/ABA] treatment

Response 4: “respond to maturation culture” has been replaced by “mature in response to maturation treatment”

Point 5: In Fig. 13, the contents of IAA, ABA and IAA:ABA ratio are shown - same as in Fig. 11, instead of Put, Spd and Spm contents!!!! This needs to be corrected, and the values for Put, Spd and Spm, for each cell line, should be presented instead!

Response 5: We have made corrections to the picture

Point 6:This is inconsistent with the data shown in Fig. 7, where it is apparent that e.g. line 1-23 yielded almost 4 SEs (PE I + PE II +PE II + ESE), with >2 PE III, or line 8-30 where >2 PE I and >1.5 ESE were obtained. Maybe these cell lines did not produce mature SEs, in which case this must be clearly denoted in the legend, as well as on the y-axis of the graph!

Response 6: The data in Figure 2 is the number of cotyledon somatic embryos, and we have supplemented the information on the y-axis of Figure 2.

Point 7: Days of culture

Response 7: “Culture of day” has been replaced by “Days of culture”

Point 8: what is PR?

Response 8: PR stands for proliferation rate, which we supplement in Figure 5.

Point 9: Abbreviations should be used consistently throughout the text and figures. So, it should be either PE or PEM in the entire document (and not both of them). Since the majority of figures and the main text contains PE (I, II, III), I suggest that the authors make appropriate corrections in Figure 5

Response 9: “PEM” has been replaced by “PE”

Point 10: What is SE? Matured SEs? Germinated SEs? Or all early stages taken together?

Response 10: “SE” has been replaced by “Cotyledonary SE”

Point 11: What is prot in (ng per g prot)?

If prot=protein, then why is starch content expressed per g protein (and not per g FW)?

If this was a mistake, and there should be FW instead, then the authors are advised to check carefully all the graphs and make corrections in both figures and the main text.

Response 11: The units of starch and soluble sugar content in Figure 8 were an error, which we have corrected in the figure and text.

Point 12: Figure 13 contains WRONG GRAPHS (!!!), showing levels of IAA, ABA instead of showing levels of Put, Spm and Spd !!!This  must be corrected!

Response 12: Figure 13 has been corrected

Point 14: Again, there is the question of whether the 'SE ability' refers to SE induction or to the production of mature SEs. The authors are referred to the document with comments concerning the title of the manuscript

Response 14: In our study, "SE ability" refers to the ability to generate mature SE. Corrections have been made.

Point 15: The article referenced as [31] is the same as that referenced as [29]!

Response 15: Done. Please see above.

Point 16: Reference(s) for the basic/modified Litvay's medium is necessary!

Response 16: Done. Please see above.

Point 17: If this is the reference for the mLV medium, it should be placed at its first mention in the Material and Methods (i.e. in the subsection 4.1.)

Response 17: Done. Please see above.

Point 18: Highlighted phrase should be deleted

Response 18: Done. Please see above.

Round 2

Reviewer 1 Report

The ms shows some improvement, but unfortunately part of the questions have not been answered. At the same time, revision has produced some new mistakes to be corrected. As I have a feeling the authors have missed some of my previous points, I’ll this time number all the issues that need to be answered / revised in the ms (not only in the letter to reviewer), before the ms can be acceptable for publication.

11)      Check the language and the text e.g. for spelling errors and extra words (as present in the r 285)

22)      To me, validation of the results means that they are confirmed in an independent experiment. In the present ms, the authors call e.g. Spm and Spd results validated although they gave in the second validation experiment the opposite results compared with the original experiment. Please find another phrasing to tell these results, both in the abstract and in the text

33)      The ms underlines correlations between SE yield and several studied factors. One of the major issues is that it is, however, not clear if these correlations were correctly calculated, as it is not described in statistical analyses. This information needs to be added !! I.e. not only to mention the program but to tell if parametric or nonparametric correlations were used. It appears that SE production data is not normally distributed (half of the lines giving zero as a result), and the number of cell lines is small - and with this kind of data it is not justified to use parametric correlation. This a very key point to address !!

44)      If pollen parents are not known, tell this in the text

55)      Clarify the subculturing regime during proliferation test; if no subculture, tell it

66)      Likewise, please say in the text what experiment were repeated for verification. And if all the factors were not measured (as one sees by comparing figures 5 and 15), tell it and give also reasoning for this choice

77)      Authors replied that they used three-month old embryogenic tissue as material – however, their text (r 385) still refers to material after two rounds of proliferation. Does this mean that during three months they subcultured their tissue only twice ?

88)      The authors revised the text by duplication results in the text – that was quite the opposite what I asked when I commented  that there is e.g. no need to tell genotypic differences in detail in the text when these are clearly seen in the figures.

99)      For genotypic differences, it is not told if the comparisons were made over both R and B lines, or separately for these two groups.  Please give this information !

110)   It would be good to know if the produced SE were able to germinate and convert into plants too ?

111)   Further, I am also missing a critical evaluation if the found markers really provide a way to select genotypes e.g. for mass-propagation prior to maturation ?? Or if the safest way is still to just test their SE production capacity ?? OR, is it more useful to use the found markers as indicators to develop the treatments along the SE propagation to find ones that result in majority of the genotypes showing indicators favourable for high embryo production ??

112)   Labels of the lines in the figure 3 have disappeared, please add them

113)   r483 is told that 1-fold content is significantly higher… cannot be, because 1-fold is the same as the one to which it is compared.

Author Response

Dear Reviewer,

Our sincere thanks to you for the time and effort that you have put into reviewing our manuscript! We found all the comments very constructive and helpful, and have revised our manuscript according to all comments. Please find, below, our point-by-point response to the comments raised.

Thank you for considering our revised manuscript!

Point 1: Check the language and the text e.g. for spelling errors and extra words (as present in the r 285)

Response 1: We have checked language and text.

Point 2: To me, validation of the results means that they are confirmed in an independent experiment. In the present ms, the authors call e.g. Spm and Spd results validated although they gave in the second validation experiment the opposite results compared with the original experiment. Please find another phrasing to tell these results, both in the abstract and in the text

Response 2: The relationship between Spd and Spm and somatic embryo yield was opposite in the two experimental results. Therefore, we speculate that the content differences of Spd and Spm are mainly affected by genotype. The link between the content of Spd and Spm and somatic embryo yield may still need a lot of experiments to verify.

Point 3: The ms underlines correlations between SE yield and several studied factors. One of the major issues is that it is, however, not clear if these correlations were correctly calculated, as it is not described in statistical analyses. This information needs to be added !! I.e. not only to mention the program but to tell if parametric or nonparametric correlations were used. It appears that SE production data is not normally distributed (half of the lines giving zero as a result), and the number of cell lines is small - and with this kind of data it is not justified to use parametric correlation. This a very key point to address !!

Response 3: We supplemented the description of correlation analysis in Methods, we used Pearson's bivariate correlation analysis method to observe the degree of association between two variables. Can only measure whether there is a relationship, and the strength/magnitude of the relationship.

Point 4: If pollen parents are not known, tell this in the text

Response 4: We have modified the method of 4.1.

Point 5: Clarify the subculturing regime during proliferation test; if no subculture, tell it

Response 5: We revised 4.2 in Methods to supplement the number of passages of embryogenic callus.

Point 6: Likewise, please say in the text what experiment were repeated for verification. And if all the factors were not measured (as one sees by comparing figures 5 and 15), tell it and give also reasoning for this choice

Response 6: We added reasons for choosing validation metrics in 2.7 and 4.3.

Point 7: Authors replied that they used three-month old embryogenic tissue as material – however, their text (r 385) still refers to material after two rounds of proliferation. Does this mean that during three months they subcultured their tissue only twice ?

Response 7: The specific information of the experimental materials we have supplemented in 4.1.

Point 8: The authors revised the text by duplication results in the text – that was quite the opposite what I asked when I commented  that there is e.g. no need to tell genotypic differences in detail in the text when these are clearly seen in the figure

Response 8: We have made changes in the paper.

Point 9: For genotypic differences, it is not told if the comparisons were made over both R and B lines, or separately for these two groups.  Please give this information !

Response 9: We compare cell lines in R and B lines separately, already supplementary information in the paper.

Point 10: It would be good to know if the produced SE were able to germinate and convert into plants too ?

Response 10: Somatic embryo germination experiments were not performed in this study.

Point 11: Further, I am also missing a critical evaluation if the found markers really provide a way to select genotypes e.g. for mass-propagation prior to maturation ?? Or if the safest way is still to just test their SE production capacity ?? OR, is it more useful to use the found markers as indicators to develop the treatments along the SE propagation to find ones that result in majority of the genotypes showing indicators favourable for high embryo production ??

Response 11: In general, our results provide a new perspective for early screening of cell lines with high SE maturation potential and provide a theoretical basis for subsequent studies based on differences in somatic embryogenesis. And using the found markers as indicators to develop the treatments along the SE propagation to find ones that result in majority of the genotypes showing indicators favourable for high embryo production. Thereby promoting the large-scale application of somatic embryogenesis technology in forest tree breeding.

Point 12: Labels of the lines in the figure 3 have disappeared, please add them

Response 12: Labels have been added in Figure 3.

Point 13: r483 is told that 1-fold content is significantly higher… cannot be, because 1-fold is the same as the one to which it is compared.

Response 13: we have modified.